METHODS

# BuDDI: *Bulk Deconvolution with Domain Invariance* to predict cell-type-specific perturbations from bulk

**Natalie R. Davidson**[1], **Fan Zhang**[1,2], **Casey S. Greene**[1]*

**1** Department of Biomedical Informatics, University of Colorado Anschutz School of Medicine, Aurora, Colorado, United States of America, **2** Department of Medicine Rheumatology, University of Colorado Anschutz School of Medicine, Aurora, Colorado, United States of America

* casey.s.greene@cuanschutz.edu

## Abstract

While single-cell experiments provide deep cellular resolution within a single sample, some single-cell experiments are inherently more challenging than bulk experiments due to dissociation difficulties, cost, or limited tissue availability. This creates a situation where we have deep cellular profiles of one sample or condition, and bulk profiles across multiple samples and conditions. To bridge this gap, we propose BuDDI (BUlk Deconvolution with Domain Invariance). BuDDI utilizes domain adaptation techniques to effectively integrate available corpora of case-control bulk and reference scRNA-seq observations to infer cell-type-specific perturbation effects. BuDDI achieves this by learning independent latent spaces within a single variational autoencoder (VAE) encompassing at least four sources of variability: 1) cell type proportion, 2) perturbation effect, 3) structured experimental variability, and 4) remaining variability. Since each latent space is encouraged to be independent, we simulate perturbation responses by independently composing each latent space to simulate cell-type-specific perturbation responses. We evaluated BuDDI's performance on simulated and real data with experimental designs of increasing complexity. We first validated that BuDDI could learn domain invariant latent spaces on data with matched samples across each source of variability. Then we validated that BuDDI could accurately predict cell-type-specific perturbation response when no single-cell perturbed profiles were used during training; instead, only bulk samples had both perturbed and non-perturbed observations. Finally, we validated BuDDI on predicting sex-specific differences, an experimental design where it is not possible to have matched samples. In each experiment, BuDDI outperformed all other comparative methods and baselines. As more reference atlases are completed, BuDDI provides a path to combine these resources with bulk-profiled treatment or disease signatures to study perturbations, sex differences, or other factors at single-cell resolution.

## Introduction

Single-cell RNA sequencing (scRNA-Seq) technologies have provided methods to interrogate how cell type proportions and cell-type-specific expression profiles vary within biological systems. In contrast, bulk RNA-Seq sequencing technologies average cell-type-specific

**Data availability statement:** All code is available on GitHub. The BuDDI model code is available at https://github.com/greenelab/buddi, and the code to recreate all analyses is available at https://github.com/greenelab/buddi_analysis. The trained models and processed data needed to recreate the analyses is available on figshare under the DOIs: https://doi.org/10.6084/m9.figshare.25564344, https://doi.org/10.6084/m9.figshare.23721336, and https://doi.org/10.6084/m9.figshare.27280626.

**Funding:** This work was supported by the National Institutes of Health (K99HG012945 and R00HG012945 to N.R.D.; R01CA237170 to C.S.G.), the Gordon and Betty Moore Foundation (GBMF 4552 to N.R.D. and C.S.G.), the Arthritis National Research Foundation Award (to F.Z.), the PhRMA Foundation (to F.Z.), and the University of Colorado Translational Research Scholars Program Award (to F.Z.). The funders had no role in study design, data collection and analysis, decision to publish, or manuscript preparation. Funding Organization Websites: National Institutes of Health: https://www.nih.gov/ Gordon and Betty Moore Foundation: https://www.moore.org/ Arthritis National Research Foundation Award: https://curearthritis.org/scientists/fan-zhang/ and https://curearthritis.org/ PhRMA Foundation: https://www.phrmafoundation.org/grants-fellowships/award-recipients/fan-zhang-phd/ and https://www.phrmafoundation.org/ University of Colorado Translational Research Scholars Program Award: https://medschool.cuanschutz.edu/program-to-advance-physician-scientists-translational-research/physician-scientist-initiatives/Early-faculty-TRSP/translational-research-scholars-2023.

**Competing interests:** The authors have declared that no competing interests exist.

differences but are easier and cheaper to perform. Due to these inherent differences, larger single-cell experiments typically provide more cell types and numbers of cells but are still lacking in the breadth of individuals, diseases, and perturbations of existing bulk RNA-Seq data. However, understanding cell-type-specific responses is key to understanding treatment response and disease etiology. For example, the method of action of traditional disease-modifying antirheumatic drugs (tDMARDs) is not well understood but is believed to target T-cells [1]. Unfortunately, there is very limited single-cell data with tDMARDs treatments. However, there are large single-cell studies measuring the arthritic synovial tissue [2,3] without tDMARDs and bulk studies that track patients before and after taking tDMARDs [1]. This pattern of missing data is not particular to arthritis and tDMARDs; it is also present in cohorts of rare diseases where the recruitment of new patients to perform single-cell sequencing is infeasible. To effectively utilize the existing large bulk studies and growing single-cell references, we need methodological advances that combine multi-condition bulk and single-condition scRNA-Seq data to estimate cell-type-specific expression profiles across the conditions observed in the bulk data. To accomplish this goal, we build on ideas from three methodological approaches: bulk deconvolution [4–14], variational autoencoder (VAE) [15] models for perturbation prediction [16–22,23], and disentanglement methods [18,24–28].

Bulk deconvolution methods unify single-cell and bulk data types by attempting to deconvolve an observed bulk expression profile as a sum of cell-type-specific expression profiles [4–14,23]. One key limitation of this deconvolution approach is that most methods assume the bulk expression profile is similar to the reference single-cell profiles. BayesPrism [13] addresses this problem using a Bayesian framework to directly account for differences between the observed bulk and single-cell data for one cell type among those with fixed profiles. We account for not only the differences between the bulk and single-cell data but additionally other sources of variation, such as sample variability and perturbation response. Furthermore, we seek to independently perturb each source of variation to simulate cell-type-, sample-, and perturbation-specific differences. We would also like our deconvolution method to be flexible and easily integrated into a larger generative model, similar in structure to Scaden, a VAE-based bulk deconvolution method [7].

There exist several generative methods to learn interpretable latent spaces that decompose the input single-cell expression profiles into relevant sources of variation. These methods can be directly trained to capture a specific source of variation [29–35] or post-hoc-interpreted after training [36–40]. Furthermore, there exist several methods to learn a latent space such that shifts within the latent space represent specific perturbation effects on an unobserved cell or cell type [4–14]. Instead of leveraging perturbation responses in other cells or cell types, we would like to leverage complex bulk expression profiles, not only cell lines or single-cell profiles, to infer the cell-type-specific perturbation response.

However, to simulate accurate perturbation responses, it is key that perturbing one latent space does not affect another latent space, i.e., changing the latent space that represents cell type proportion should only affect the variability related to cell type proportions, and not other sources of variability related to the sample identity or sequencing technology. This concept is related to domain invariance, where latent representations are invariant to changes in a domain. For example, if the prediction task is to count the number of cells in an image, and the trained model can do so accurately on images with varying brightness, coloration, and resolution, the model is considered invariant to these differences in the brightness, coloration, and resolution domains. One difference between our proposal and typical domain invariance approaches is that our main goal is not for our method to be invariant of unseen domains but invariant to observed domains within our dataset of interest. In our case, we would like to model each latent representation to be independent of one another, which could also be

phrased as having latent representations that are disentangled. Specifically, this means that changes in one latent representation are independent of changes in all other latent representations. This framework can be used to learn classifiers invariant to a specific confounding factor [25,28] or to analyze the latent spaces to interrogate the sources of variability within the data [24,26,27]. Our use case requires the generative aspect of the model to predict cell-type-specific perturbation effects similar to MichiGAN [18], except we will infer the perturbation response from bulk data, not single-cell.

BuDDI combines strategies to learn domain-invariant representations that capture cell type proportions, perturbation effects, and experimental variability. BuDDI not only learns interpretable latent representations to understand the data better but can also compose changes in each latent space to predict cell-type-specific perturbation responses.

## Results

### The model structure of BuDDI

We implement BuDDI as a Variational Autoencoder (VAE) [15], since it is a generative model, straightforward to train, and its modular structure. Briefly, a VAE is a model trained to re-generate the original input data after it has been compressed through a bottleneck layer. The bottleneck layer is typically much smaller than the input data and is reflective of the main sources of variation within the data, which we will refer to as a latent representation of the data. BuDDI extends the VAE structure by having more than one latent representation, and all but one latent representation is directed to capture a specific source of variability - the remaining latent representation, termed the 'slack', captures unexplained variability. BuDDI's VAE structure (Fig 1) reflects the belief that our observed gene expression data is generated from at least four sources of variability: sample or technical variability ($z_e$), condition-specific variability ($z_p$), differences in cell type proportion ($z_y$), and other sources of noise ($z_x$). To ensure each latent space is specific to its source of variability, an auxiliary loss is added to BuDDI to predict the labels related to the sample, technology, condition, and cell type proportion. Since BuDDI learns from bulk and single-cell RNA-Seq data, the cell type proportions are not always known; therefore, $z_y$ is trained semi-supervised, and $z_e$ and $z_p$ are trained fully supervised. $z_x$ is unrestricted but is the same dimensionality as $z_e$ and $z_p$. A more detailed description of the training procedure and model is given in Methods.

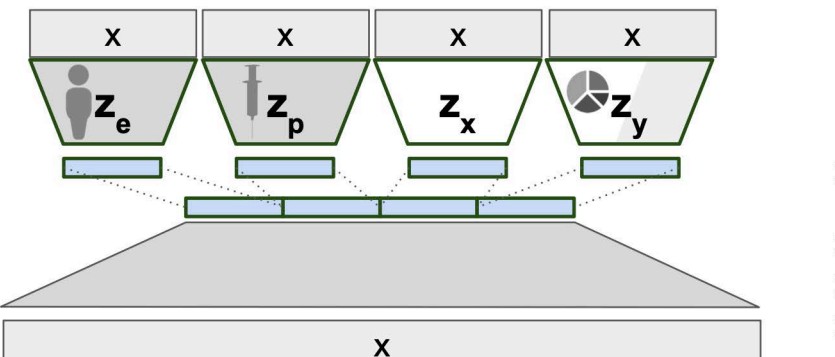

**Fig 1. VAE structure of BuDDI.** X is our bulk or pseudobulk. We apply an auxiliary loss on each latent code for them to encapsulate a specific source of variability. Since our model is generative, we can later sample from each latent space to simulate experimental changes to our input expression profile. To simulate cell-type-specific effects, we can sample a cell type proportion where the cell type of interest is the predominant cell type. The person, syringe, and pie chart icons were obtained from openclipart [41–43].

BuDDI utilizes the generative model structure introduced in DIVA [28], a method to identify disentangled latent representations in cellular images. Similarly, BuDDI treats each of these sources of variability as specific and invariant domains. Domain invariance is key to BuDDI learning cell-type-specific perturbation effects since we can independently learn representations for the perturbation and cell type and compose them together to learn a cell-type-specific effect.

While the generative structure of BuDDI encourages each latent space to be invariant, real biological data is unlikely to have training data with independent sources of variability. Specifically, cell type proportions are likely dependent on the sample or perturbation status. To break this dependence, we simulate pseudobulk data used in training to have random cell type proportions. This allows us to break the dependence between cell type proportions and the other sources of variation. The approach assumes the observed expression data is sufficiently independent for the remaining latent spaces to learn descriptive and domain-invariant representations. In the following sections, we evaluate this assumption, finding that BuDDI works on data with increasing levels of interdependence across the latent representation. Firstly, we validate BuDDI on the simplest experimental design using only pseudobulks, where we have matched samples across each source of variability. Next, in a more realistic setting, we still use pseudobulks but now have no matched samples between bulk and single-cell. Finally, we test BuDDI on real single-cell and bulk data from Tabula Muris Senis [44,45], where there are no matched samples across any source of variation.

## BuDDI learns descriptive and domain-invariant latent representations

To validate that BuDDI works as expected, we first tested the simplest experimental design, where we have matched observations across each source of variability. We used a dataset created by Kang et al. [46] of peripheral blood mononuclear cells from two of the eight lupus patients with matched samples that either had interferon-Beta stimulation or no stimulation. To simulate bulk samples, we omitted cell type proportions from half of the pseudobulks during training. An overview of the data included in our experimental design is shown in Fig 2A.

After training BuDDI, we measured the extent of domain invariance across latent spaces. We compared the predictive accuracy of each latent space in predicting its intended and unintended targets on a held-out test set. This is similar to the Separated Attribute Predictability (SAP) score [47], except we compare distinct latent spaces to one another instead of an individual latent dimension. Each latent space approximated domain invariance: the accuracy of each latent space to predict its intended source of variability was significantly higher than a mismatched source of variability (Fig 2B). This indicated that each latent space was specific to only its intended target, not targets described by another latent space. Furthermore, we observed that each latent space was not only relatively accurate in predicting its intended target but generally accurate; each latent space was predictive of its intended source of variability with a very high F1 score (>0.9). We also observed that BuDDI can learn the cell type proportions of the pseudobulk data accurately, as shown by the strong correspondence between ground truth and predicted cell type proportions (Fig 2C).

After quantitative evaluation, we also qualitatively evaluated the specificity of each latent space. We observed that the first two principal components (PCs) divide each latent space by its target value, demonstrated in the plots along the diagonal of Fig 2D. Furthermore, along the off-diagonal, the non-target sources of variability are well mixed. This indicated that most of the variance in the latent spaces specifically captures the target source of variability. Finally, we qualitatively examined whether any variability remained unexplained by the other latent spaces. Specifically, we evaluated the slack latent space, which is designed to capture any residual variability not captured by the supervised or semi-supervised latent spaces. We

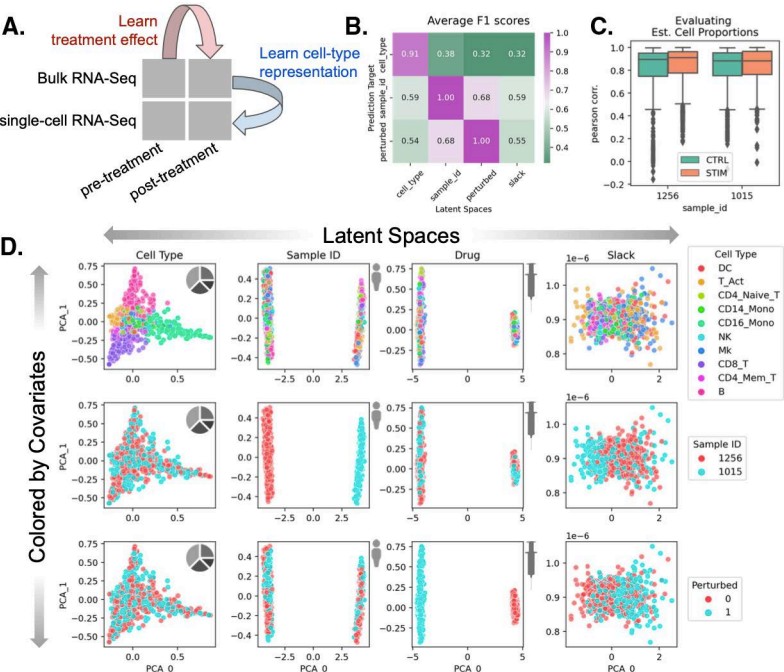

**Fig 2. Evaluation of BuDDI on pseudobulk data with matched samples across each source of variability.** (A) Panel A depicts a schematic of the experimental design. (B) Panel B depicts a heatmap of the average F1 score using each latent space to predict each source of variability. A high F1 score along the matched latent space and source of variability, and a low F1 score where the latent space does not match the source of variability is a measure of disentanglement across the latent spaces. (C) Panel C shows the performance of BuDDI at predicting the cell type proportions. (D) Panel D visualizes the first two principal components (PCs) of each latent space (columns) and colors them by different sources of variation (rows). The person, syringe, and pie chart icons were obtained from openclipart [41–43].

observe that in the slack latent space, each target is well mixed, indicating that it is not capturing variability from explicitly modeled sources. We also observe a lack of clear structure in the slack latent space, indicating that there is little remaining structured variability to be explained by the slack.

## BuDDI accurately predicts cell-type-specific perturbation response

After validating that BuDDI learns specific latent space representations, we examined the extent to which BuDDI predicts cell-type-specific perturbation responses when perturbation measurements are only available in bulk data. Again, we used the data from Kang et al. [46] to generate our simulated data, except used all eight available samples. To make the bulk data more comparable with actual data, we simulated realistic cell type proportions that were again omitted during training. Furthermore, to examine the method's ability to identify a cell-type-specific effect and not simply a global shift, we only use stimulated CD14 monocytes for simulation (Fig 3A).

First, we determined whether or not BuDDI could capture the perturbation response in our dataset when not explicitly modeled. We trained an augmented version of BuDDI (BuDDI-noPert), where we removed the perturbation latent space. The BuDDI-noPert slack latent space captured the perturbation response (Fig 3B). Once the perturbation space was reintroduced, the slack space no longer separated the samples by perturbation status (S1A Fig; the slack space was not strongly predictive of the perturbation status; mean F1 score: 0.52). Additionally, the latent

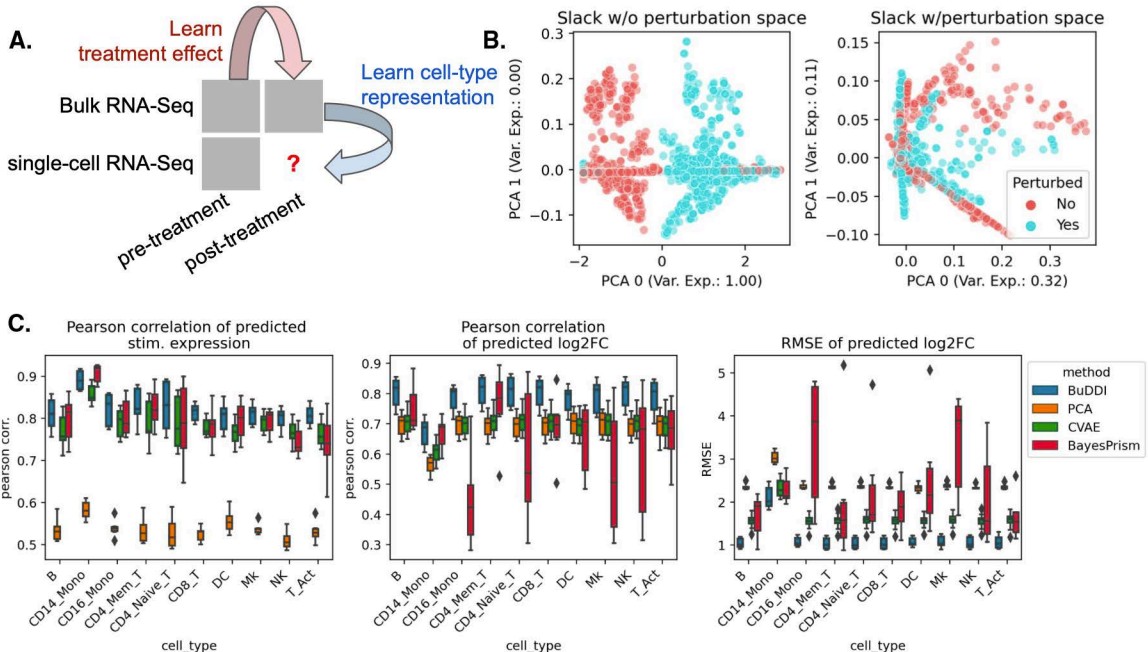

**Fig 3. Evaluation of BuDDI on cell-type-specific perturbation simulation.** BuDDI on pseudobulk data with matched samples across each source of variability. (A) Panel A depicts a schematic of the experimental design; we no longer include the single-cell perturbation response during training. (B) Panel B depicts the slack space when training BuDDI without (left) and with the perturbation latent space (right). Here we observe that when we train BuDDI without the perturbation space, the slack space picks up the perturbation response. This effect is greatly diminished once we include the perturbation latent space. (C) Panel C depicts the performance of BuDDI, PCA, BayesPrism, and CVAE in predicting the cell-type-specific expression and log2 fold change. In this experiment, only CD14 monocytes are stimulated. To evaluate the model variability of BuDDI and CVAE, each model was trained and evaluated three independent times and is included in (C).

spaces were still generally predictive of and specific to their specific source of variation, although as expected, performance was degraded in comparison with the experiment where paired samples were supplied across each source of variability (S1A–S1C Fig).

Next, we identified if BuDDI could predict the expression and effect size of the perturbation for each cell type. We compared BuDDI against PCA with latent space projections, BayesPrism [13], and a conditional VAE (CVAE) [48]. To get cell-type-specific expressions for PCA and CVAE, we used the pseudobulks generated primarily from one cell type, then applied the perturbation. For PCA, we learned a sample-specific linear translation to simulate the perturbation. For CVAE, the perturbation and sample IDs were included in the conditions, so we only had to change the condition status in the CVAE on the pseudobulks with primarily one cell type to simulate a cell-type-specific perturbation effect. We evaluated each method on pseudobulks generated from held-out single-cell RNA-Seq profiles. Full details of the experimental design are given in Methods. Across all metrics and cell types, BuDDI outperformed all other methods (Fig 3C). Since our experimental design only perturbs CD14 monocytes, it is unsurprising that we see performance degradation in that cell type; however, BuDDI still outperforms all other methods and maintains a relatively high Pearson correlation for the predicted stimulated expression (mean > 0.8) and log2 fold change (mean > 0.65). We then examined if performance was degraded in more lowly expressed genes. We observed that CVAE performance increases for more highly expressed genes (S1D Fig). BuDDI also performs better with higher levels of expression, but the performance increase was not as drastic. BuDDI's performance was comparable to PCA for lowly expressed genes and comparable to

CVAE on highly expressed genes, with BuDDI outperforming all models when considering all levels of expression (S1D Fig). To further validate BuDDI, we assessed its performance across a broader range of cell types and perturbations using sci-Plex3 data [49]. BuDDI was applied to computationally mixed single-cell data from three cell lines (A549, K562, MCF7) and five drugs, each with distinct mechanisms of action. We found that, across all cell lines and drugs, BuDDI consistently demonstrated either superior or comparable false positive rates and area under the PR curve (S2 Fig).

## BuDDI accurately identifies cell-type-specific sex differences

Finally, we examined the extent that BuDDI predicted cell-type-specific sex differences in the Tabula Muris Senis dataset [44,45]. Tabula Muris Senis consists of male and female mice's bulk and single-cell expression data in several organs. We restricted our analysis to the liver, a sexually dimorphic organ. The challenge of this dataset is that there are no matched samples across any source of variability. There were no technical replicates for any samples nor matched bulk and single-cell samples. Furthermore, we do not have matched perturbation effects to examine sex differences because each mouse was either male or female. This experimental design implies that each source of variability is highly entangled with each other. We evaluated predictions using a held-out single-cell female mouse sample (Fig 4A).

First we examined whether or not BuDDI separated the sources of variability in this highly correlated dataset. We visually found that each latent space was specific to its target source of variability (Figs 4C and S3). Importantly, we observed a clear separation between the cell type and the sex, the two latent factors required predict cell-type-specific sex differences (Fig 4C). However, some entanglement remained between the slack and cell type latent spaces (S3 Fig).

Next, we aimed to predict genes with the largest sex differences in each cell type. In contrast to experiments using perturbation data, obtaining matching expression data across sexes is impossible. Because it is not possible to validate predictions by predicting each sample's exact gene expression value for each sample since we have no ground truth, we identified the top genes predicted to have the largest difference in expression between the sexes. In addition to CVAE, BayesPrism, and PCA, we also compare against: random, a baseline of the shuffled predicted values; zero, a baseline of the majority label (0); and bulk, a baseline of the differentially expressed genes between the bulk samples. The bulk baseline represents the global shift in expression; therefore, outperforming the bulk baseline indicates that the model identifies cell-type-specific differences. We compared our results against two validation sets. The first set is the differentially expressed genes between the single female and male mice provided by Tabula Muris Senis [44,45]. We provide full details of the data processing and differential expression pipeline in Methods. The second validation set is from an independent study of sex differences using single-nucleus RNA-Seq data [53]. We included this secondary study since it has more biological replicates and is from a complementary sequencing platform.

BuDDI outperforms all other methods and baselines in each cell type, including the bulk baseline, indicating that BuDDI can identify cell-type-specific sex differences beyond a global shift in expression (Figs 4D and S4). PCA with a latent transformation is the only method to outperform the bulk expression in only one cell type, hepatic stellate cells. In all other cell types, PCA and CVAE perform similarly and are better than random but are significantly outperformed by BuDDI.

## BuDDI predicts cell-type-specific pathway responses to immunosuppressive drug

After validating that BuDDI identified cell-type-specific sex differences in the mouse liver, we applied BuDDI to real bulk data perturbed by the IL-6R inhibitor Tocilizumab. Tocilizumab

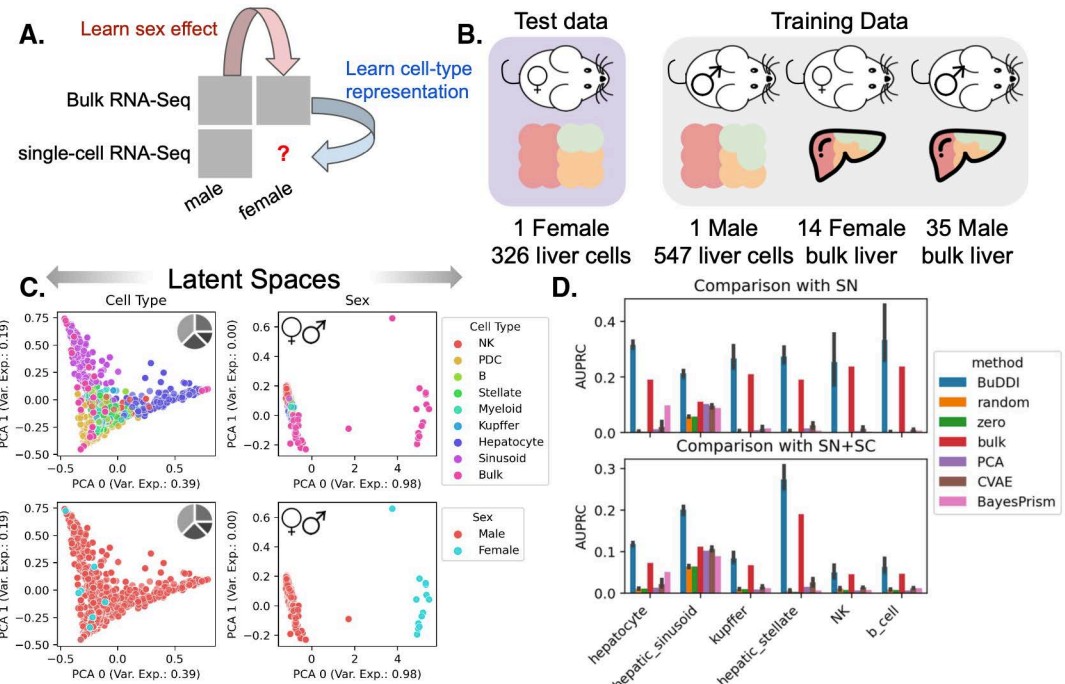

**Fig 4. Evaluation of BuDDI to predict cell-type-specific differences in the mouse liver.** (A, B) Panels A and B depict a schematic of the experimental design and data used for training and evaluation. (C) Panel C depicts the cell type and sex latent spaces colored by either the most abundant cell type or sex. The data in the PCA plots contains all pseudobulk and real bulk training data. (D) Panel D depicts the area under the Precision-Recall curve in predicting the differential gene between the sexes for each cell type. (D), top, uses differentially expressed genes identified by an independent single-nucleus experiment analyzing sex-specific differences in the liver. (D), bottom, uses the union of differentially expressed genes from the afore-mentioned single-nucleus experiment and the Tabula Muris Senis [44,45] single-cell experiment. Bar height represents the mean area under the precision-recall curve (AUPRC) and the black lines indicate the 95% confidence interval. To consider the model variability of BuDDI and CVAE, each model was trained and evaluated three independent times. The pie chart, mouse, male, and female icons were obtained from openclipart [43,50–52].

inhibits IL-6, a pro-inflammatory cytokine, from binding to IL-6R to induce an anti-inflammatory effect [54–58]. There is currently no single-cell data of synovial tissue pre- and post-treatment, therefore, only traditional differential expression analyses using bulk RNA-Seq data are possible. However, the bulk analyses may be confounded by changes in cell type proportions between conditions or cannot detect expression changes in low-proportion cell types. BuDDI overcomes this gap by integrating bulk and single-cell data to infer the missing cell-type-specific responses. We trained BuDDI on untreated single-cell synovial tissue [3] and bulk pre- and post-treatment synovial tissue from individuals with rheumatoid arthritis [57].

To examine whether or not BuDDI could identify higher resolution pathway changes than using bulk RNA-Seq alone, we generated pre- and post-treatment pseudobulks with a uniform cell type proportion. We use uniform cell type proportions to 1) identify pathway changes in rarer cell types and 2) control for changes in cell type proportions due to treatment. The differential analysis revealed that real bulks and BuDDI-generated pseudobulks were enriched for the inflammatory response and multiple cytokine-related pathways (Fig 5A). This was expected since these are broader pathways likely to affect multiple cell types. When we looked more specifically at the inflammation pathway across cell-type-specific expression changes inferred by BuDDI, we observed that each cell type was enriched for the inflammation path-way (Fig 5B). We then focused on the more specific IL-6-related pathways. We found that

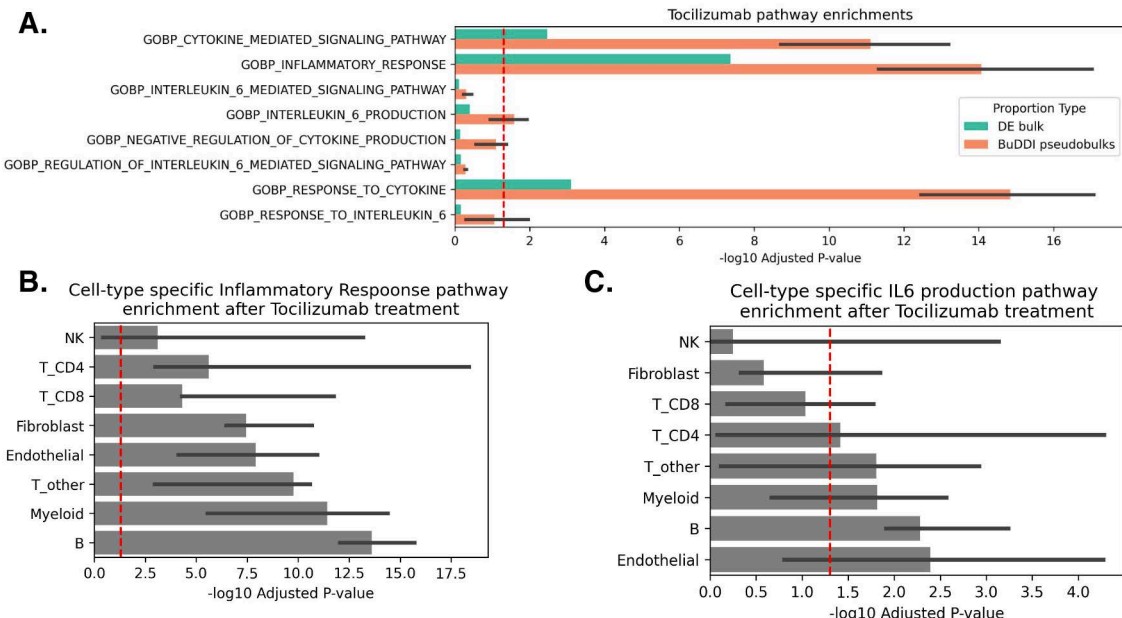

**Fig 5. BuDDI prediction of pathway changes induced by Tocilizumab treatment.** (A) Panel A depicts the enrichment of Tocilizumab-relevant pathways in the top 500 genes for real bulk and BuDDI-generated pseudobulk data across three independently trained BuDDI models (the thick bar is the median, and the thin bars are the lowest and highest observed -log10 (p-adjusted). The BuDDI-generated pseudobulks were simulated with uniform cell type proportions to control for rare cell types and differences in cell type proportions across treatments. (B) Panel B depicts the cell type specific enrichment for the Inflammation pathway inferred by BuDDI. (C) Panel C depicts the cell type specific enrichment for the IL-6 production pathway inferred by BuDDI.

the BuDDI-generated pseudobulks were more enriched for IL-6-specific pathways than the real bulks ([Fig 5A]). To explain this difference, we inspected the cell-type-specific pathway differences. We observed that not all cell types were affected equally by Tocilizumab treatment. Instead, it primarily affected Endothelial, B, Myeloid, CD4 T, and remaining non-CD8 T cell types ([Fig 5C]). This finding aligns with the current understanding of cell-type-specific expression of IL-6 and IL-6R, the target of Tocilizumab. IL-6 is produced by several cell types, including T cells and endothelial cells [59]. While IL-6R is not expressed on endothelial cells and only on a subset of T-cells, these cell types can still respond to IL-6 using trans signaling [60,61].

## Discussion

We introduce BuDDI, a method to learn cell-type-specific perturbation responses using reference single-cell and multi-condition bulk data. BuDDI learns latent representations specific to a single source of variation and independent of all other sources of variation. This model design enables BuDDI to individually perturb one or more latent spaces and compose them to simulate cell-type-specific perturbations. In most experimental designs, it is impossible to have data that has matched samples across all sources of variability. We successively evaluated BuDDI on increasingly entangled data, moving from data that had all, some, and then no matched samples across the sources of variability. BuDDI's unique model design enables researchers interrogate the sources of variability within their data. The model's slack space, $z_x$, captures remaining variability that was not directly modeled, allowing researchers to identify unaccounted confounders. We also found that BuDDI outperforms or matches all competitor models and baselines in each simulated and real experiment. While BuDDI

is more computationally demanding than PCA and CVAE, it is much less demanding than BayesPrism. To generate S2 Fig, PCA and CVAE take under four minutes to train and apply to a new dataset, BuDDI takes roughly 25 minutes to train and under 4 for application, BayesPrism takes over two hours for building and applying their model to a subset (<50%) of the entire dataset.

BuDDI can be tuned in different ways. There is an inherent tradeoff between the accuracy of latent representation and the reconstruction, which leads to significant degradation of the cell type proportion estimator when the experimental design has more entangled sources of variability (S3 Fig). In our evaluations, we optimized the reconstruction accuracy of BuDDI to predict cell-type-specific perturbation response. Depending upon the use case, the end-user can specifically train BuDDI to have a better cell type proportion estimator, but at the cost of reconstruction accuracy. Furthermore, while we use three classifiers in most of our evaluations, BuDDI can currently support the use of 2 or 3 classifiers and can be extended to add or remove classifiers. It is also possible to remove all classifiers and assign sources of variations to specific nodes in the latent representation post-hoc. However, these subsets of nodes will not necessarily be disentangled, implying that directly changing the value in a single node may affect more than one source of variability. The beta-VAE is a VAE variant that encourages disentangled latent representations but still requires post-hoc data interpretation. BuDDI provides an easy-to-use method to have both transparent and disentangled latent representations. One drawback of BuDDI is that it does not test for the number of relevant sources of variation that best describe the data. In our experiments, three classifiers best represented the independent sources of variation, but this is an experimental design choice that should be made by the end-user. The complexity of the model and the amount of available data should be considered by end-users, especially when considering the risk of overfitting when using a limited amount of bulk data. One potential extension of the model could involve incorporating dropout hyperparameters to mitigate this risk. However, it is essential to carefully evaluate held-out data to ensure overfitting does not occur.

Another caveat to using BuDDI, is that BuDDI assumes that given all sample information, such as cell type proportion, sample ID, perturbation status, sequencing differences, etc., a normal pseudobulk and normal bulk should have highly correlated expression. However, this assumption is violated when cell types are missing in single-cell data but present in bulk. While we do learn a technology-specific transformation of the data, this transformation will inherently become entangled with the cell type proportion latent space and degrade our predicted cell-type-specific perturbation responses. The correlated normal pseudobulk and normal bulk assumption is also broken when the reference single-cell data is more variable in expression than a normal pseudobulk. This could be caused by stress induced by cell dissociation. This will only be an issue if the technology-specific transformation is correlated with another modeled source of variability, so if the same stress pathway was activated in the perturbation, our perturbation and technology latent spaces would now be entangled. When it is suspected that the variability between normal pseudobulk and bulks may be inappropriately matched, the user can quantify the level of entanglement by using the SAP score analysis depicted in S1A Fig to provide guidance on whether the application of BuDDI is appropriate.

While we evaluated BuDDI on expression data, this implementation is conceptually extendable to other data types. The approach can be applied to other data modalities as long as it is possible to generate augmented training data that separates the cell-type-specific signal from the other sources of variation. Three natural extensions for BuDDI can be used to unite across 1) single-cell and bulk ATAC-Seq to get cell-type specific peaks, 2) cell-free DNA and ATAC-Seq to detect cell- or tissue-specific fragments, and 3) single-cell RNA-Seq and spatial

RNA-Seq to deconvolve spatial spots. Furthermore, other than cell type proportion, we have currently implemented BuDDI to represent sources of variability only as discrete values. Conceptually, BuDDI could model continuous sources of variability, such as age, perturbation time, or drug concentration. The structure of BuDDI is also very modular, easily allowing users to extend BuDDI to include additional contraints, such as a clustering constraint within the sample latent space. This allows the user to identify, for example, tumor subtypes that are independent of confounders captured in other latent spaces.

BuDDI provides a methodological solution to a missing data pattern that is common in genomic analyses of publicly available data. Without needing to sequence more, BuDDI can leverage one technologies' depth in its cellular profiles with another's breadth in the heterogeneity of profiles. BuDDI has several potential use cases, such as providing a way to analyze tissues whose cells are difficult to dissociate at a single-cell resolution, to leverage difficult-to-obtain data from patients with rare diseases, or to re-analyze the tens of thousands of heterogeneous existing bulk samples. BuDDI strives to make the most out of existing bulk datasets in the era of large-scale single-cell reference atlases.

## Methods

### BuDDI model description

BuDDI extends the VAE framework [15] and uses a similar conceptual structure as DIVA [28]. The entire VAE structure attempts to find a latent representation (z) that is likely to recon-struct the original data (x). The goal is to maximize the marginal likelihood [15,62].

$$p_\theta(x) = \int p_\theta(x|z) p_\theta(z) dz \tag{1}$$

$p_\theta(x|z)$ is the decoder and uses a neural network to learn the parameters $\theta$, where given z we reconstruct x. Unfortunately, learning $p_\theta(x)$ is intractable, since it requires integrating over all possible latent representations $z$. Instead, we estimate it by learning a lower bound to $p_\theta(x)$, by learning an approximate posterior $q_\phi(z|x)$. $q_\phi(z|x)$ is our encoder, where $\phi$ are learned parameters of the encoder neural network. We can rewrite $p_\theta(x)$ as

$$log p_\theta(x) = \mathbb{E}_{q_\phi(z|x)}\left[log\left(\frac{p_\theta(x,z)}{q_\phi(z|x)}\right)\right] + \mathbb{E}_{q_\phi(z|x)}\left[log\left(\frac{q_\phi(z|x)}{p_\theta(z|x)}\right)\right] \tag{2}$$

$$= L_{\theta,\phi}(x) + D_{\mathbb{KL}}\left(q_\phi(z|x)||p_\theta(z|x)\right) \tag{3}$$

Since $D_{\mathbb{KL}}\left(q_\phi(z|x)||p_\theta(z|x)\right)$ is non-negative, $L_{\theta,\phi}(x)$ is a lower bound on $log p_\theta(x)$. Now we learn parameters to maximize $L_{\theta,\phi}(x)$, which can be rewritten as

$$L_{\theta,\phi}(x) = \mathbb{E}_{q_\phi(z|x)}\left[log\left(p_\theta(x|z)\right)\right] - \beta D_{\mathbb{KL}}\nu\left(q_\phi(z|x)||p_\theta(z)\right) \tag{4}$$

where $\beta$ is a weighting term to constrain the amount of variability that can be explained by the latent space [63]. Unlike a VAE with a single latent space ($z$), DIVA and BuDDI learn independent latent spaces to capture different sources of variability (experimental $z_e$, per-turbation $z_p$, and remaining variability $z_x$ This is done through learning separate encoders, $q_{\phi_e}(z_e|x)$, $q_{\phi_p}(z_p|x)$, and $q_{\phi_x}(z_x|x)$, and a single decoder. To capture variability due to cell type proportions, we directly append the observed cell type proportion to the latent space when it is available or use a predicted cell type proportion from an auxiliary predictor when the cell type proportion is not available. This implies that $z_y \approx y$, instead of being predictive

of $y$ as done in the other latent spaces. The auxiliary predictor takes the gene expression $x$ as input and predicts the cell type proportion, $y$, and it's weights are only updated when the cell type proportions are known. This is how BuDDI is able to predict the cell type proportions in a semi-supervised fashion. The loss without the auxiliary proportion loss, but including the additional latent spaces is the following:

$$L_{\theta,\phi}\left(x\right) = \mathbb{E}_{q_{\phi_e}(z_e|x)q_{\phi_p}(z_p|x)q_{\phi_x}(z_x|x)q_{\phi_y}(z_y|x)}\Big[log\left(p_{\theta}\left(x\,|\,z_e,z_p,z_x,y\right)\right)\Big] - \beta_e D_{\mathbb{KL}}\left(q_{\phi_e}\left(z_e\,|\,x\right)\|\,p_{\theta}\left(z_e\right)\right)$$
$$- \beta_p D_{\mathbb{KL}}\left(q_p\left(z_p\,|\,x\right)\|\,p_{\theta}\left(z_p\right)\right) - \beta_x D_{\mathbb{KL}}\left(q_{\phi_x}\left(z_x\,|\,x\right)\|\,p_{\theta}\left(z_x\right)\right) \tag{5}$$

A more detailed derivation of $L_{\theta,\phi}\left(x\right)$ can be found in the original DIVA manuscript [28]. Unlike DIVA, we do not use conditional priors to separate the latent spaces from one another and instead only use auxiliary classifiers on the experiment and perturbation specific latent spaces, $q_{\omega_e}\left(e\,|\,z_e\right)$ and $q_{\omega_p}\left(p\,|\,z_p\right)$, to constrain the latent spaces to their intended source of variability. The full loss is.

$$L_{BuDDI}\left(x\right) = L_{\theta,\phi}\left(x\right) + \alpha_e \mathbb{E}_{q_{\phi_e}(z_e|x)}\Big[log\left(q_{\omega_e}\left(e\,|\,z_e\right)\right)\Big] + \alpha_p \mathbb{E}_{q_{\phi_p}(z_p|x)}\nu\Big[log\left(q_{\omega_p}\left(p\,|\,z_p\right)\right)\Big]$$
$$+ \alpha_y \mathbb{E}\Big[log\left(p_{\theta_y}\left(y\,|\,x\right)\right)\Big] \tag{6}$$

A detailed diagram of the BuDDI implementation is provided in S4 Fig.

## BuDDI training and implementation details

In generating the pseudobulks used for testing and training, cells were divided into two even sets stratified by each source of variation: perturbation status, cell type, and sample ID. Therefore, pseudobulks used in training will not have any cells seen in testing. BuDDI was implemented in Keras version 2.12.0, and was trained using the Adam optimizer [64], with a learning rate of 0.005. The non-slack $\beta$ terms are always set to 100 and $\beta_x$ is set to 0.1. This parameter choice encourages the non-slack latent representations to be biased towards fully capturing the source of variability, since a larger $\beta$ term creates a stronger bottleneck on the latent representation and encourages stronger disentanglement within the latent space [63]. The number of epochs [50, 100, 200] and the classifier weights [10, 100, 1000, 10000, 100000] were identified using a grid search of all combinations. We minimized reconstruction loss and maximized the Spearman correlation of the true and estimated cell type proportions on a training validation set, which is 20% of the training set held out during training. After the initial set of classifier weights was identified, the latent spaces were visually inspected and individual classifier weights were increased by a factor of 10 if further disentanglement of an individual latent space was needed. For all models, excluding the sci-Plex3 model, we used 64 dimensions for each latent representation and a batch size of 500. We used internal dimensions of 512 and 256 for the cell type proportion predictor. We used a single 512-dimensional dense layer for the perturbation and experimental predictors. The full structure of our model, including the dimensions of each hidden layer is provided in S5 Fig. All model choices that were not searched over were chosen early in model design on simulated data experiments and kept consistent throughout all evaluations. All hyperparameter choices and ranges searched are provided in S5 Table.

To train BuDDI cell type proportions in a semi-supervised manner, we created two separate encoder models with shared weights. When the cell type proportions are not known, the cell type proportion predictor weights are not updated, and its predictions are used in the latent space during training. When the cell type proportions are known, the cell type

proportion predictor weights are updated, but the predictions are not used in the latent space. Instead, the true value is directly input into the latent space during training. This is depicted as two separate model diagrams in S5 Fig. During training, BuDDI switches between the supervised and unsupervised models within each epoch. In both cases, the auxiliary classifiers for predicting the sources of variation, excluding the cell type proportions, are always supervised, and their weights are updated throughout the entire epoch.

The structure of each latent space is identical to one another, with two hidden layers of dimensions 512 and 256. In all experiments, we have two latent spaces representing experiment-specific variability, $z_e$, one that is predictive of the sample ID and the other that predicts whether the data comes from a pseudobulk sample or a real bulk sample. For the BuDDI-noPert experiment, the perturbation latent space $z_x$ is excluded from the entire model.

## BuDDI simulation of perturbation response

BuDDI learns a separate latent space for each source of variability, allowing us to modify a specific latent space to simulate a change related to that latent space. To do this, we use our training data to sample latent codes that predict a specific source of variability. We can perturb a single latent space or several latent spaces and combine them to produce the desired latent representation. We use a y with the highest cell type proportion for the cell type of interest to generate a cell-type-specific perturbation effect. We will combine this with latent codes related to unperturbed and perturbed samples. Combining these two latent codes with the remaining latent codes relevant to the experiment, we compared the gene expression differences between the perturbed and unperturbed samples for a specific cell type. Depending on the desired analysis, the additional latent spaces could be sampled randomly or specific to a sample of interest. For the Kang et al. [46] data with matched samples, we sampled latent codes specific to each sample. We jointly sampled the latent slack, sample, perturbation, and bulk codes for the tocilizumab and sex-dependent liver analysis. When the latent spaces were observed to have high amounts of independence between them, each latent space could be sampled more independently. Conversely, if high dependence between latent spaces is observed, it is recommended to jointly sample the latent spaces that are not directly relevant to the perturbation of interest. We also note that by using our sampling and data generation approach, we will generate several examples of a perturbed version of a single, unperturbed sample. To collapse our simulated samples back to a single sample, we aggregate by the median for each gene.

## CVAE model description

The CVAE [48] learns a latent representation conditioned on specific variables; in our case, we implemented a CVAE conditioned on the sample ID, perturbation status, and whether the input data is pseudobulk or a real bulk. The CVAE differs from a VAE in its implementation by appending a 1-hot-encoded vector representing the sources of variation to the input to both the encoder and the decoder. After training, new data is generated by changing the appended vector to represent the perturbation of interest. However, unlike BuDDI the vector representing the source of variation cannot be trained in a semi-supervised manner. Therefore, it is impossible to learn a model that is conditional on the cell type proportions and the perturbation status since we only have perturbed observations from the bulk data, which has no cell type proportion estimate. To get around this limitation, we instead learn a latent space that captures the cell type proportions and is independent of all other sources of variation. This enables us to calculate cell-type-specific perturbation changes by sampling from regions in the latent space specific to a cell type, then appending our latent code that represents our perturbation of interest.

The CVAE was implemented in Keras. For consistency, we maintained the same latent code dimension as BuDDI and the same dimension of encoder and decoder layers. We also used the same optimizer, ADAM, with a learning rate of 0.005. The $\beta$ term was set to 1 in all experiments. $\beta$ values were grid searched [0.1, 1, 10] to minimize the reconstruction error and identify a latent space that was predictive of the cell type proportions.

## PCA model description

PCA was used to learn a low-dimensional data representation. We then learned a linear transformation between the perturbed and non-perturbed samples in the low-dimensional representation. To learn a cell-type-specific perturbation response, we used pseudobulks with a cell type proportion where the cell type of interest was the majority cell type. Next, we summed its low-dimensional representation with the perturbation vector and projected the sample back into the original dimensionality of the data. Since we had matched samples for the Kang et al. [46] data, we also learned a sample translation vector and the perturbation vector to simulate a sample-, cell-type-, and perturbation-specific effect. The number of latent dimensions used for PCA was 20, which explained >90% of the variability in both datasets.

## Data processing

The single-cell data used in each experiment was processed using scanpy [65]. For all experiments, the cell type labels were taken from the original manuscript. The Kang et al. [46] analysis data was downloaded from SeuratData [66] and converted to h5ad format for downstream processing in scanpy. In the Kang et al. [46] analysis, we removed outlier cells with less than 500 or more than 2500 genes expressed. We removed genes expressed in less than five cells. The total number of cells used by cell type and sample are shown in S1 Table.

The data for the sci-Plex3 analysis were downloaded from NCBI's GEO (GSM4150378). In this data, we focused on five drugs (Fedratinib, Tanespimycin, Trametinib, Triamcinolone Acetonide, and Trichostatin A) that were previously reported to have an observed response in each of the three cell lines (A549, MCF7, and K562). We removed cells with more than 30% mitochondrial reads and less than 500 genes expressed. Additionally, we kept cells that were treated with no drug or treated at 1000nM. We only kept genes with at least one count in at least 50 cells in each cell line. After filtering, we are left with 13917 cells (MCF7: 7858, K562: 3132, A549: 2927) and 8022 genes across the three cell lines.

The data for the sex-specific liver differences were downloaded from the Tabula Muris Senis [44,45] project (https://figshare.com/articles/dataset/Processed_files_to_use_with_scanpy_/8273102/2), hosted by FigShare [https://doi.org/10.6084/m9.figshare.8273102.v2]. Due to a low number of cells and expressed genes in the liver dataset, we could only analyze one male and one female mouse sample. Two male mice samples had a sufficient number of cells for each cell type, but we restricted our analysis to post-pubescent mice (3 months or older), which resulted in the filtering of one of the male mice. Furthermore, hepatic stellate cells were very rarely observed (<27 cells per sample, 3.25 on average) and therefore combined with endothelial cells of the hepatic sinusoid, a more abundant cell type with a similar expression profile. We did not filter cells, but we removed genes expressed in less than three cells. S2 Table provides the counts of cells by sample.

The bulk liver data was downloaded from Gene Expression Omnibus under accession ID GSE132040. We filtered samples that were less than three months old. The total number of samples by age and sex are provided in S3 Table. We did not perform additional count processing on the single-cell data before pseudobulk generation for each dataset. Additional processing was only done for identifying differentially expressed genes in the single-cell data. Raw counts were used for differential expression analysis of the bulk data, as needed for pyDESeq2 [67].

The single-cell data used to predict a cell-type-specific Tocilizumab effect was downloaded from the manuscript-provided synapse link (https://doi.org/10.7303/syn52297840) with further help from the author [3]. The original data files were converted to the h5ad format for scanpy. Cells with fewer than 500 genes and genes expressed in fewer than 100 cells were removed from the analysis. The total number of cells used by cell type for each sample is provided in S4 Table. Only samples with sufficient expression were used in the downstream analyses (S8 Fig). The bulk data used to predict the cell-type-specific Tocilizumab effect originated from the Rivellese et al. [57] dataset. BuDDI was trained using samples treated with Ritixumab, Tocilizumab, and untreated samples. To estimate pathway enrichment, we only used samples with paired pre- and post-Tocilizumab effects. This includes both responders and non-responders. Due to differences in the gene expression counts between the pseudobulk and real bulk data, we performed 90th-percentile normalization between the pseudobulks and real bulks by multiplying the pseudobulk counts by the ratio of 90th percentiles between the two types of bulk data.

## Pseudobulk generation

After processing the data, as described in the Data processing section, we performed a 50/50 split of the cells, stratified by sample and cell type. This ensured we did not observe any pseudobulks with shared cells between the training and testing sets. To create the pseudobulks, we summed over sampled cells from each individual dependent upon a specific cell type proportion. We generated three types of cell type proportions: random, cell-type-specific, and realistic. Random proportions were sampled from a lognormal distribution, with a mean of 5 and a variance uniformly sampled between 1 and 3. All proportions were scaled to sum to 1. The cell-type-specific proportions were generated by first creating a vector of the length of cell types where the cell type of interest had a proportion of $1 - \left( \left( \# celltypes \right) * 0.01 \right)$, and the remaining cell types had a proportion of 0.01. Lognormal noise with mean 0 and variance 1 was added to the cell type proportions and then rescaled such that they sum to 1. Suppose the new cell type proportion did not have a Pearson correlation coefficient >0.95 with the original cell type proportion vector before the noise was added. In that case, noise vector was discarded, and a new one was sampled. The realistic cell type proportion estimator calculated the sample-specific cell type proportion observed from the single-cell data. Noise was added in the same way as for the random cell type proportions. After the cell type proportions were sampled, we sampled a total of 5000 cells dependent upon the cell type proportion and sum over the counts to generate the pseudobulk values. S6 Fig depicts the generated pseudobulks with each type of sampled proportion.

## Differential expression of single-cell and bulk data

Differential single-cell expression was done using scanpy [65] and pyDESeq2 [67]. We first generated cell-type-specific pseudobulks, generating ten samples and 30 cells sampled per cell type. Using these pseudobulks, we used pyDESeq2 to identify the genes that were differentially expressed between the sexes for each cell type. For the bulk and pseudobulk pyDESeq2 analyses, genes with a mean expression across all samples <1 were removed from the analysis. We considered genes with adjusted p-value <0.01 as differentially expressed for all downstream analyses. The single-nucleus differentially expressed genes were taken from [53].

## Pseudobulk normalization

After the pseudobulk data was generated, it was uniformly processed for each experiment and model. First, we identified 7000 genes that form the union between CIBERSORTx-identified

signature genes [4] and the genes we calculated to have the highest coefficient of variance. These genes were highly overlapping (S7 Fig). Next, we MinMax scaled the gene expression. Since gene counts typically have long-tailed expression profiles, we clipped the expression at the 90th quantile before scaling.

## Predicting source of variability using each latent space

To predict each source of variability, we used a Naive Bayes classifier. We reported the average F1 score on a held-out test set of 10% of the data. We performed this classification task 30 times for each model. To take into account the variability of BuDDI, we independently trained three separate BuDDI models and averaged their performance.

## Pathway enrichment

All pathway scores were estimated using the method Enrichr from the package GSEApy [68]. The GO Biological Process gene sets used in the Tocilizumab analysis were downloaded from www.gsea-msigdb.org. We used the median rank difference between treated and untreated simulated data. Since we were interested in the negative regulation of IL-6-related pathways, we ranked the genes from negative to positive and took the top 500 to calculate pathway enrichment. The background geneset consisted of all genes used in training BuDDI. The pathways were chosen to depict those most related to Tocilizumab treatment effects.

## Evaluation of models applied to sci-Plex3 data

We generated 15 tests to evaluate each model's performance on sci-Plex3 data. Each test corresponds to a single drug and cell line combination. Perturbed pseudobulks were generated from a single perturbed cell line and the two remaining unperturbed cell lines. The non-perturbed pseudobulks were generated from all non-perturbed cell lines. Reference single cell expression was provided only from non-perturbed single cells. Each method was then evaluated for how well it reproduced unseen perturbed cell-type specific expression. All cells were divided into test and training cells, so no cells used in pseudobulk generation were also used for evaluation. True positives were the top differentially expressed genes for each cell line. The metrics used to evaluate how well each model correctly identifies differentially expressed genes were the area under the precision-recall curve for identifying the top 10 or 50 differentially expressed genes. To evaluate the false-positive rate of each method, we identified how many genes were identified as differentially expressed between the non-perturbed cell lines after simulating the cell-line-specific perturbation effect. A gene was identified as differentially expressed if the Bonferroni-corrected p-value was <0.05 after applying a t-test of ten simulated perturbed and non-perturbed samples. For BayesPrism, a subsample of ten perturbed and non-perturbed samples was used for evaluation. BuDDI, PCA, and CVAE were trained independently three times for evaluation.

## Evaluation of genes predicted to be sex-dependent

Since we could not have matched samples from different sexes, we could not directly compare sample- and cell-type-specific changes in gene expression due to sex. Instead, we predicted the genes most affected by sex differences for each cell type. We compared the simulated male and female gene expression for each model for each cell type. We then reported the median rank difference between male and female simulated data. To calculate the area under the precision-recall curve (AUPRC), we used the absolute value of the median rank difference. Our true values were either from an independent single-nucleus experiment [53] that identified sex-dependent genes, or from the genes identified as sex-dependent from the Tabula

Muris Senis data [44,45] used to generate the pseudobulks. The comparative baselines were 1) random: shuffled ranks; 2) zero: a predictor that only reported zero, the majority label; and 3) bulk: the sex-dependent genes identified by analyzing the bulk Tabula Muris Senis data.

## Supporting information

**S1 Fig. Latent space analysis of BuDDI on Kang et al. [46] data set with an experimental design where bulk samples are correlated with the sample IDs and perturbation status. Panel a** depicts that average F1 score of each latent space to predict each source of variation. Midpoint coloration is the average across all observed F1 scores. **Panel b** compares the performance of BuDDI, CIBERSORTx, and BayesPrism, in estimating the cell type proportions. **Panel c** depicts each of BuDDI's latent spaces, colored by source of variation. **Panel d** depicts the Pearson correlation of the simulated perturbation expression, stratified by expression level. (PDF)

**S2 Fig. Estimating cell line specific perturbation effects in sci-Plex3 data. Panel a** Schematic of the experimental design. 15 tests were used to evaluate each model's performance on sci-Plex3 data. Each test corresponds to a single drug and cell line combination. More specifically, each perturbed pseudobulk consists of a single cell line perturbed by a single drug, all other cell lines are unperturbed. **Panel b** Performance evaluation of each model on 15 sci-Plex3 tests, corresponding to specific drug and cell line combinations. True positives were the top differentially expressed genes per cell line, measured using area under the precision-recall curve (AUPRC) for the top 10 and 50 differentially expressed genes. False-positive rates were assessed by identifying genes falsely classified as differentially expressed between non-perturbed cell lines, using a Bonferroni-corrected p-value <0.05. BuDDI, PCA, and CVAE were independently trained three times, while BayesPrism used subsampled perturbed and non-perturbed data. The syringe icon was obtained from openclipart [42]. (PDF)

**S3 Fig. Latent space analysis of BuDDI on Tabula Muris Senis dataset.** Each column is a latent space and each row is colored by a source of variation. The second row is colored by sample ID, but due to the number of bulk samples, we omit the sample ID legend. (PDF)

**S4 Fig. ROC and PR curves for predicting differentially expressed genes between sexes in hepatocytes using BuDDI.** Top row uses the differential expressed genes form an independent single-nucleus experiment [53] as the ground truth, bottom row uses the union of the single-nucleus and our calculated single-cell results from Tabula Muris Senis [44,45] as the ground truth. (PDF)

**S5 Fig. BuDDI model overview for the supervised (top) and unsupervised (bottom) models.** The red box highlights the true or estimated cell type proportions used in BuDDI. (PDF)

**S6 Fig. Pseudobulk data generated and colored by source of variation.** Our generated data shows independence between, each source of variation, including cell type proportion. (PDF)

**S7 Fig. Overlap of top coefficient of variation genes and CIBERSORTx signature genes used in the Kang et al. [46] (left) and sex-dependent liver (right) analyses.** (PDF)

**S8 Fig. Log total counts for each single-cell synovium sample from Zhang et al. [3].** Only samples with sufficient expression were used in our analysis, this includes samples 421, 436, 458, 460, 462, 475, 515, and 542.
(PDF)

**S1 Table. Number of cells by cell type and by sample ID in the Kang et al. [46] dataset after filtering.**
(PDF)

**S2 Table. Number of cells by sample ID and cell type after filtering and before combining the two cell types "endothelial cell of hepatic sinusoid" and "duct epithelial cell".**
(PDF)

**S3 Table. Number of bulk liver samples used in analysis by sample ID and age.**
(PDF)

**S4 Table. Number of cells by sample ID and cell type from Zhang et al. [3].**
(PDF)

**S5 Table. Hyperparameters for each trained BuDDI model.**
(PDF)

## Acknowledgments

The authors acknowledge Dr. Faisal S. Alquaddoomi for his valuable discussions on methodology and assistance with software implementation.

## Author contributions

**Conceptualization:** Natalie R. Davidson, Casey S. Greene.

**Data curation:** Natalie R. Davidson, Fan Zhang.

**Formal analysis:** Natalie R. Davidson.

**Funding acquisition:** Natalie R. Davidson, Casey S. Greene.

**Investigation:** Natalie R. Davidson, Casey S. Greene.

**Methodology:** Natalie R. Davidson, Casey S. Greene.

**Project administration:** Casey S. Greene.

**Resources:** Fan Zhang, Casey S. Greene.

**Software:** Natalie R. Davidson.

**Supervision:** Fan Zhang, Casey S. Greene.

**Validation:** Natalie R. Davidson.

**Visualization:** Natalie R. Davidson.

**Writing – original draft:** Natalie R. Davidson, Casey S. Greene.

**Writing – review & editing:** Natalie R. Davidson, Fan Zhang, Casey S. Greene.

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
