## [Decision Letter · Decision Letter 0]

8 Aug 2024

Dear Dr. Greene,

Thank you very much for submitting your manuscript "BuDDI: Bulk Deconvolution with Domain Invariance to predict cell-type-specific perturbations from bulk" for consideration at PLOS Computational Biology. As with all papers reviewed by the journal, your manuscript was reviewed by members of the editorial board and by several independent reviewers. The reviewers appreciated the attention to an important topic. Based on the reviews, we are likely to accept this manuscript for publication, providing that you modify the manuscript according to the review recommendations.

Sincerely,

Jacob G. Scott, MD

Academic Editor

PLOS Computational Biology

Shihua Zhang

Section Editor

PLOS Computational Biology

Reviewer's Responses to Questions

**Comments to the Authors:**

Reviewer #1: In this manuscript, Davidson and colleagues present a new computational method to decovolve cell-type perturbations. Their method, BuDDI, uses domain invariance tools to leverage bulk sequencing and single-cell sequencing data to predict cell-type expression shifts during treatment/perturbation.

This is a very well-written manuscript – it is logical and easy to follow. Moreover, I commend the authors for the careful documentation of their code, which they have already posted to github. Overall, I think this tool addresses a pressing need in the field, and I suspect it will be adopted by many labs.

I have only minor comments:

1. The authors use the terminology ‘slack space’ in several portions of the paper. While those with machine learning expertise may understand what the authors intend, I would encourage the authors to a sentence in the results section (during the portion covering Figure 2) to introduce the term more clearly.

2. The discussion is short – I think the manuscript would benefit from more details about how the authors expect BuDDI to be used and expanded. Moreover, the authors mention that BuDDI may be applicable to other data types. I think more speculation here would be welcome. For example, I could certainly see application of BuDDI to methylation or proteomic data. Moreover, do the authors see a future for extending the BuDDI framework to spatial data types?

3. Finally, it would be helpful to get a sense of how computationally demanding BuDDI is. How well does BuDDI scale to large datasets, and does it compare favorably to BayesPrism/PCA/CVAE/etc…?

Reviewer #2: The authors proposed a novel method for integrating single-cell RNA sequencing (scRNA-seq) reference data with corpora of case-control bulk RNA-seq data (named BuDDI) to infer cell-type-specific perturbation effects. BuDDI should help the community bridge this integration gap by learning domain-invariant latent representations within a variational autoencoder (VAE) framework, disentangling sources of variation such as cell type proportion, perturbation effect, and experimental variability. The authors validate BuDDI on simulated and real datasets, demonstrating its ability to predict cell-type-specific perturbation responses and sex-specific differences.

BuDDI addresses the fundamental problem of integrating single-cell and bulk RNA-seq data using domain adaptation techniques and disentangled latent representations within a VAE framework. The results show that BuDDI generally outperforms existing methods in various experimental settings, demonstrating its effectiveness in predicting cell-type-specific responses. Considering the vast amount of existing data, BuDDI has been tested on a few data (i.e., sex-specific differences and drug response in a single context).

In general, the paper is well-written, and the achieved results are promising and well-presented. However, the authors should address some concerns before a possible publication in PLoS Computational Biology.

Major comments

- The authors should consider applying BuDDI to a wider range of real-world datasets, including different tissues, diseases, and perturbations. This would provide a more comprehensive assessment of BuDDI’s generalisation abilities and robustness.

- The manuscript could benefit from a more intuitive explanation of the VAE framework and domain adaptation techniques. Additional visualisations or diagrams could help readers understand the model's architecture and how it disentangles sources of variation.

- The authors should benchmark BuDDI against a wider range of existing deconvolution and integration methods. This would provide a more thorough assessment of BuDDI's performance relative to the state-of-the-art.

- Considering the model complexity, BuDDI might overfit especially when dealing with limited real-world data. The authors should discuss the potential for overfitting and present strategies, such as regularisation techniques, to mitigate this issue.

The mathematical derivation of the modified ELBO lacks many steps. In addition, the authors should better explain why and how they decided to introduce three classifiers.

Minor comments

- Clarify the definition of "domain invariance": The term "domain invariance" is used throughout the manuscript, but its precise definition is not explicitly stated. A clear definition would help the readers understand the concept and its relevance.

- The authors mention using grid search for hyperparameter tuning (just for some hyperparameters) but do not provide details on the search space or the specific metrics used for optimisation. More information on this process would be helpful for reproducibility. Moreover, the authors decide to manually set other hyperparameters (e.g., learning rate and beta values). How did they select these values? What about the number of layers and neurons per layer?

- The authors should discuss BuDDI’s limitations, such as its reliance on reference scRNA-seq data and potential challenges in dealing with highly heterogeneous tissues.

- The authors should also outline potential future directions for the method, such as extending it to other omics or incorporating additional sources of variation.

Reviewer #3: The study titled “BuDDI: Bulk Deconvolution with Domain Invariance to predict cell-type-specific perturbations from bulk” by Davidson et al. aims to utilize deep representation learning through variational autoencoders specifically designed to identify components associated with predefined sources of variation enabling the transfer of trained network to single-cell setting for prediction of transcriptional perturbations.

The is an important and non-trivial problem where the availability of bulk sequencing data allows for investigations of across-sample variations whereas single-cell data is generally limited to a handful of samples albeit with increased sensitivity to detect sub-clonal populations of cells.

Overall, I think the authors do a good job of evaluating the proposed model in various settings both qualitatively and quantitatively. Couple minor points remains to be addressed.

- For the perturbation prediction and sampling from the latent space, how is the stochasticity handled? Meaning that for a single pseudo-bulk input sample multiple predictions can be made since sampling from the latent space, are the predictions presented based on mean/median aggregate of multiple samples? This should be clarified in the text.

- The encoder architecture is interesting, and I think should be discussed/contrasted more. Vanilla version of an encoder specifically, given enough capacity, should also capture distinct latent spaces. Effectively a single slack space could capture the sources of variation albeit with different latent dimensions.

- Discussion should be expanded including potential limitations of modeling pseudo-bulk from single-cell data to integrate bulk sequencing data with single-cell. Specifically, how would the increased heterogeneity in single-cell data effect model performance and the relevance of pseudo-bulk data to true bulk data.

- More details should be given regarding the parameter tuning such as batch size, number of layers, latent dimension, β, size etc.

- Please provide the explained variance ratios in figure 4c.

- Please specify whether the PCA of latent space corresponding to the cell-type (figure 4c) is showing the held-out female single-cell data.

- Supp Figure 1, legend should be bold for panels (Panel b,c,d) for uniformity.

**Have the authors made all data and (if applicable) computational code underlying the findings in their manuscript fully available?**

Reviewer #1: Yes

Reviewer #2: Yes

Reviewer #3: Yes

PLOS authors have the option to publish the peer review history of their article (what does this mean? ). If published, this will include your full peer review and any attached files.

**Do you want your identity to be public for this peer review?** For information about this choice, including consent withdrawal, please see our Privacy Policy .

Reviewer #1: No

Reviewer #2: **Yes: ** Andrea Tangherloni

Reviewer #3: No

Figure Files:

While revising your submission, please upload your figure files to the Preflight Analysis and Conversion Engine (PACE) digital diagnostic tool, https://pacev2.apexcovantage.com . PACE helps ensure that figures meet PLOS requirements. To use PACE, you must first register as a user. Then, login and navigate to the UPLOAD tab, where you will find detailed instructions on how to use the tool. If you encounter any issues or have any questions when using PACE, please email us at figures@plos.org.

Data Requirements:

Please note that, as a condition of publication, PLOS' data policy requires that you make available all data used to draw the conclusions outlined in your manuscript. Data must be deposited in an appropriate repository, included within the body of the manuscript, or uploaded as supporting information. This includes all numerical values that were used to generate graphs, histograms etc.. For an example in PLOS Biology see here: http://www.plosbiology.org/article/info%3Adoi%2F10.1371%2Fjournal.pbio.1001908#s5 .

Reproducibility:

References:

---

## [Decision Letter · Decision Letter 1]

20 Dec 2024

Dear Dr. Greene,

We are pleased to inform you that your manuscript 'BuDDI:Bulk Deconvolution with Domain Invariance to predict cell-type-specific perturbations from bulk' has been provisionally accepted for publication in PLOS Computational Biology.

Best regards,

Jacob Scott, MD

Academic Editor

PLOS Computational Biology

Shihua Zhang

Section Editor

PLOS Computational Biology

Kudos.

Reviewer's Responses to Questions

**Comments to the Authors:**

Reviewer #1: The authors have done a great job of addressing my concerns. I think it should be accepted as is.

Reviewer #2: The authors carefully replied to all my critiques and suggestions point-by-point, improving the quality and readability of the previous version of the manuscript. Thus, I think that the current version of the manuscript could be suitable for publication in PLoS Computational biology.

Reviewer #3: I do not have any additional comments, the authors did a good job on responding and improving the manuscript/study from the initial review stage

**Have the authors made all data and (if applicable) computational code underlying the findings in their manuscript fully available?**

Reviewer #1: Yes

Reviewer #2: Yes

Reviewer #3: Yes

PLOS authors have the option to publish the peer review history of their article (what does this mean? ). If published, this will include your full peer review and any attached files.

**Do you want your identity to be public for this peer review?** For information about this choice, including consent withdrawal, please see our Privacy Policy .

Reviewer #1: **Yes: ** Anand G. Patel

Reviewer #2: **Yes: ** Andrea Tangherloni

Reviewer #3: **Yes: ** Arda Durmaz

---

## [Editor Report · Acceptance letter]

PCOMPBIOL-D-24-01042R1

BuDDI:Bulk Deconvolution with Domain Invariance to predict cell-type-specific perturbations from bulk

Dear Dr Greene,

I am pleased to inform you that your manuscript has been formally accepted for publication in PLOS Computational Biology. Your manuscript is now with our production department and you will be notified of the publication date in due course.

With kind regards,

Anita Estes
